# The Expression Levels of MicroRNAs Differentially Expressed in Sudden Sensorineural Hearing Loss Patients’ Serum Are Unchanged for up to 12 Months after Hearing Loss Onset

**DOI:** 10.3390/ijms24087307

**Published:** 2023-04-15

**Authors:** Reyhaneh Abgoon, Printha Wijesinghe, Cathie Garnis, Desmond A. Nunez

**Affiliations:** 1Division of Otolaryngology, Department of Surgery, University of British Columbia, Vancouver, BC V5Z 1M9, Canada; 2Vancouver Coastal Health Research Institute, Vancouver, BC V5Z 1M9, Canada; 3Department of Integrative Oncology, British Columbia Cancer Research Centre, Vancouver, BC V5Z 1L3, Canada; 4Division of Otolaryngology-Head & Neck Surgery, Vancouver General Hospital, Vancouver, BC V57 1M9, Canada

**Keywords:** sudden sensorineural hearing loss, microRNAs, serum

## Abstract

Sudden sensorineural hearing loss (SSNHL) is an acquired idiopathic hearing loss. Serum levels of small, non-coding RNAs and microRNAs (miRNAs) miR-195-5p/-132-3p/-30a-3p/-128-3p/-140-3p/-186-5p/-375-3p/-590-5p are differentially expressed in SSNHL patients within 28 days of hearing loss onset. This study determines if these changes persist by comparing the serum miRNA expression profile of SSNHL patients within 1 month of hearing loss onset with that of patients 3–12 months after hearing loss onset. We collected serum from consenting adult SSNHL patients at presentation or during clinic follow-up. We matched patient samples drawn 3–12 months after hearing loss onset (delayed group, *n* = 9 patients) by age and sex to samples drawn from patients within 28 days of hearing loss onset (immediate group, *n* = 14 patients). We compared the real-time PCR-determined expression levels of the target miRNAs between the two groups. We calculated the air conduction pure-tone-averaged (PTA) audiometric thresholds in affected ears at the initial and final follow-up visits. We undertook inter-group comparisons of hearing outcome status and initial and final PTA audiometric thresholds. There was no significant inter-group difference in miRNA expression level, hearing recovery status and initial and final affected ear PTA audiometric thresholds.

## 1. Introduction

MicroRNAs (miRNAs) are short (approximately 23 nucleotides) non-coding RNA strands that regulate gene expression. RNA polymerase II transcribes these nucleotides to create initial miRNAs in the nucleus (pre-miRNAs). Pre-miRNAs are precursor miRNAs with a hairpin structure that are derived by RNA polymerase III transcription and DGCR8 protein cleavage before transfer to the cytoplasm by exportin-5 (RNA binding protein). The RNA-induced silencing complex (RISC) is formed in the cytoplasm from loading one arm of the miRNA duplex resulting from the cleavage of pre-miRNAs by Dicer into Argonaute protein. The RISC binds to complementary sequences on target messenger RNA (mRNA), to achieve mRNA translation inhibition and mRNA degradation. MiRNAs thus control target gene expression at the post-transcriptional stage [1] based on the miRNA’s 7-nucleotide seed region targeting of specific mRNAs [2]. Currently, more than 1000 human miRNAs have been identified, and miRNAs are thought to regulate 30% of genes [3]. MiRNAs play a role in the control of numerous signalling pathways, including those involved in organogenesis, hematopoiesis, differentiation, cell proliferation, apoptosis, and fat metabolism [4].

MiRNAs have been identified in a wide range of bodily fluids, such as serum, plasma, urine, saliva, and other bodily fluids [5,6,7]; they are reliable markers of a variety of diseases [8]. There is evidence that they may play a role in cell-to-cell communication and that they can be exported from one cell and recognized, taken up, and used by another, drawing parallels with the endocrine system [9,10,11,12]. Unlike conventional RNA molecules, circulating miRNAs are highly stable and resistant to RNase activity, as well as extreme pH and temperature [2,13]. Because ribonucleases are found in body fluids, it is thought that miRNAs are protected from degradation by being packaged in lipid vesicles (microvesicles and exosomes), by forming complexes with RNA binding protein, or both [9,14,15]. More importantly, they are easily detectable in serum and plasma, and their expression patterns have been found to correlate positively with a variety of disease conditions. Their expression can reveal details about disease stages. MiRNAs are emerging as independent prognostic biomarkers for a number of diseases; it has recently been found that they can function as prognostic markers in cardiovascular disease and cancer [16]. Disease conditions involving non-cardiovascular organs also alter microRNA expression in peripheral blood. As a result, the detection and assessment of circulating miRNA expression levels offer one of the best ways to study diseases in inaccessible anatomical sites [17] since they can be identified and examined non-invasively. MiRNAs are also being investigated as a potential new class of therapeutic agents [18].

Changes in the circulating levels of miRNAs have been identified in acquired sensorineural hearing loss [19]. A better understanding of miRNA-regulated signaling pathways may facilitate the development of new hearing loss therapies [20]. Changes in microRNA expression levels with resulting upregulation of target pro-apoptotic genes is a suggested pathogenetic mechanism in presbycusis [21].

Sudden sensorineural hearing loss (SSNHL) is a type of acquired idiopathic hearing loss that develops within 72 h [22]. SSNHL has been described as having peak incidence between the ages of 43 and 53 years, with males and females being equally affected [23]. Recent large surveys have reported incidence rates of 27 to 60 per 100,000 people in the United States and Japan, respectively [24,25]. A small male predominance was seen in the United States study, but there was a 3:1 female predominance in the Japanese data. SSNHL primarily presents as a unilateral hearing loss, which is frequently accompanied by tinnitus, aural pressure, and occasionally, dizziness, nausea, and vomiting [26].

There is early evidence of miRNA involvement in SSNHL [19,27]. Li et al. [27] proposed that miRNAs hsa-miR-34a/548n/15a/143/23a/210/18b regulated target genes critical to the onset of SSNHL. Ha et al. [17] and Nunez et al. [19] found corroborating evidence for differentially expressed miRNAs (DEMs) in the blood of SSNHL patients. Furthermore, Ha et al. [17] reported that miR-15a/-18b/-183/-210 expression level in plasma drawn within 2 weeks of sudden hearing loss onset predicted the likelihood of hearing recovery. 

The reversibility of the identified DEM findings in SSNHL patients has not been studied. Hence, this study determines and compares the serum expression level of previously identified DEMs in SSNHL patients within 1 month of hearing loss onset with that of patients 3–12 months post-hearing loss onset.

## 2. Results

Nine SSNHL patients whose serum samples were classified, based on the time of collection (Table 1), as being in the delayed group were matched by age and sex with 14 SSNHL patients, from whom serum samples were obtained within 1 month of hearing loss (immediate group). 

The mean ages ± standard deviation (SD) of the patients in the immediate and delayed groups, 58.07 ± 16.95 and 59.56 ± 15.86 years, respectively, were similar. The male:female ratios in the immediate and delayed sample patient groups, 7:7 and 5:4, respectively, were similar (Table 2). Overall, 50% of patients in the delayed group had symptoms of dizziness, while no immediate group patients were similarly affected (Chi-squared test, *p* = 0.11). The initial median PTA audiometric threshold in the affected ears of immediate group patients, 66.9 (inter-quartile range IQR, 42.8) dB, was not statistically different from the 76.3 (IQR, 32.9) dB median PTA audiometric threshold recorded in the delayed group (Mann–Whitney U-test, *p* = 0.33). There was no significant inter-group difference in the proportion of patients experiencing hearing recovery, six of fourteen (42.9%) in the immediate group and five of nine (55.6%) in the delayed group. The final median PTA-affected ear audiometric threshold in immediate and delayed group patients, 63.3 (IQR, 42.9) dB and 67.5 (IQR, 68.1) dB, respectively, were also not significantly different. Both groups were thus similarly matched for presenting hearing loss severity, the proportion of patients with dizziness, and the rate of hearing recovery (Table 2).

The median expression levels ∆Ct (IQR) of miRs -128-3p/-132-3p/-375-3p/-590-5p/-30a-3p/-140-3p/-186-5p and -195-5p were 2.68 (1.12), 3.29 (1.86), −1.66 (1.54), 4.34 (2.85), 4.88 (1.87), 2.37 (2), 0.70 (1.07), 3.38 (0.97) in immediate group patients and 2.92 (1.48), 3.80 (1.38), −0.59 (1.06), 5.96 (3.05), 4.96 (1.65), 2.96 (2.08), 0.68 (2.63), 4.35 (2.77) in delayed group patients, respectively, were not statistically different (*p* > 0.005 Bonferroni-corrected Mann–Whitney U-test) (Figure 1).

## 3. Discussion

The expression levels of eight mature miRNAs found to be differentially expressed in the serum of SSNHL patients within 28 days of the onset of hearing loss compared to normal hearing healthy control individuals [19] were selected for further study. The expression levels of these eight miRNAs in serum drawn from SSNHL patients 3–12 months after SSNHL onset (delayed) were similar to those in the serum of SSNHL patients obtained within 28 days of hearing loss onset (immediate) (*p* > 0.05, Mann–Whitney U-test). Furthermore, hearing recovery as well as final PTA thresholds of the affected ears were not statistically significantly different between immediate and delayed SSNHL patient groups. The stability of these miRNA findings over time suggests that these miRNAs have translational potential as SSNHL disease biomarkers. 

The spiral modiolar artery (SMA) is, to the best of our knowledge, the only artery that supplies blood to the cochlea. This artery ascends through the modiolus and is primarily divided into two parallel capillary networks of the stria vascularis and the spiral ligament in the cochlear lateral wall [28]. The stria vascularis has a denser capillary network. Because the SMA is the only artery that supplies blood to the cochlea, SMA blockage is difficult to compensate for and results in pathological damage to the cochlea’s microcirculation [29,30]. SMA blood flow is critical in maintaining normal hearing [31,32,33,34,35]. Ischemia/hypoxia secondary to cochlear arterial thrombosis is one of the predominant proposed pathogenetic mechanisms in SSNHL. Evidence supporting this mechanism in SSNHL includes the clinical efficacy of hyperbaric oxygen therapy in reversing the associated hearing loss [36] and the predominance of target mRNAs of differentially expressed miRNAs in SSNHL patients in the PI3K/AKT signaling pathway [37]. Moreover, ischemia/hypoxia contributes to reactive oxygen species (ROS) generation, which is associated with several types of SNHL [38]. 

MiRNAs that are abundantly found in the brain play critical roles in the development and function of neuronal networks, including neurogenesis, synaptogenesis, and morphogenesis regulation [39,40]. Of the eight miRNAs tested in this study, miRs -128-3p and -140-3p have been identified as brain-enriched miRNAs (Shao et al., 2010; Adlakha and Saini, 2014). MiR-128-3p’s role in vascular smooth muscle cells (VSMCs), such as cardiomyocytes [41] and myoblasts [42], has been established. VSMCs are important cellular components of arteries, and they play a crucial role in vessel homeostasis by maintaining vascular wall tone and integrity [43,44]. VSMCs have phenotypic plasticity, which means they can change from a contractile/nonproliferating to a migratory/proliferating phenotype in response to extracellular stimuli or environmental cues [45]. A recent study has identified that miR-128-3p is significantly downregulated in VSMCs subjected to hypoxia. The overexpression of miR-128-3p decreases VSMCs proliferation and migration and helps VSCMs maintain a contractile phenotype [46]. In contrast, miR-375-3p overexpression has been implicated in carotid artery stenosis by promoting VSMC proliferation and migration [47]. MiR-375-3p expression was also found to be downregulated in several neural injury models, including cerebral ischemia/reperfusion injury (I/RI) models [48,49,50]. In our previous study, miR-128-3p was found to be significantly upregulated, and miR-375-3p downregulated in the serum of SSNHL patients compared to normal hearing control participants [9]. Therefore, the de-regulation of miRs -128-3p and -375-3p identified in SSNHL patients is likely to result in SMA VSMCs maintaining their contractile phenotype and exacerbating the adverse effects of arterial occlusion. However, their potential role in the pathogenesis of SSNHL requires further investigation.

MiR-140-3p, another brain-enriched miRNA, has been demonstrated to orchestrate neuroinflammation and neuron apoptosis [51]. MiR-140 suppresses inflammation by inhibiting NF-κB signaling in myocardial tissues in I/RI [52]. Most importantly, some researchers have found that hypoxia is related to miR-140-3p expression, and hypoxia-inducible factor -1 alpha (*HIF-1α*) is a target of miR-140-3p [53,54]. Similar to miR-140-3p, miR-186-5p is associated with hypoxia. In the in vitro analyses of primary human dermal microvascular endothelial cells, miR-186-5p downregulation is triggered by hypoxia via HIF-1α activation [55]. Additionally, miR-30a-3p, which belongs to the miR-30 family, has been reported to be downregulated across several types of malignancies and hypoxic conditions [56,57]. This study suggests that the statistically significant upregulation of miRs -140-3p and -186-5p and the downregulation of miR-30a-3p identified in SSNHL patients’ serum [9] persists for up to 12 months after the onset of hearing loss. The de-regulation of miRs -140-3p, -186-5p, and -30a-3p identified in the serum of immediate and delayed SSNHL patients may reflect the joint occurrence of a hypoxia mitigating response in the case of miR-30a-3p and an initiating or potentiating response in the case of miR-140-3p and 186-5p.

The nervous system has been shown to contain large amounts of miR-195 and miR-132 [58]. MiRs -195 and -132 target mRNAs are primarily enriched in cellular signaling pathways, including PI3K/AKT, Ras, and MAPK [19]. MAPK activation has been linked to other SSNHL syndromes, including ototoxic- and age-related hearing loss (ARHL) [59]. The brain-derived neurotrophic factor gene (BDNF) is a strong biological target of miR-132-3p [9]. Neurotrophic factors generated by the organ of Corti hair and supporting cells, in particular BDNF and neurotrophin-3, support the survival of spiral ganglion neurons in vitro [60]. Importantly, Mullen et al. [61] identified that BDNF-induced neurite formation was mediated through Ras/p38 and PI3K/AKT signaling in neonatal rat spiral ganglia explants. MiR-132 and BDNF are part of complex overlapping feedback loops involving not only signaling cascades but other miRNAs and transcription factors [62]. Similar to miR-132-3p, miR-195-5p binds directly to the 3′- untranslated region (UTR) of BDNF mRNA and inhibits BDNF protein production [63]. Moreover, miR-195-5p belongs to the miR-15 family that targets outer-hair-cell-specific prestin (*Slc26a5*) and antiapoptotic B-cell lymphoma 2 (*BCL2*) *genes* [37,64]. There is also evidence that miR-132 upregulation causes dopaminergic neuronal death via the SIRT1/P53 pathway [65]. Both miRs -132-3p and -195-5p were significantly deregulated in the serum of SSNHL patients [9] and may contribute to spiral ganglion neuron and hair cell loss by suppressing BDNF levels and increasing apoptosis. 

Additionally, miR-132 expression is upregulated by Toll-like receptors (TLR) in inflammation, which in turn exerts an inhibitory effect on inflammatory processes [66]. It downregulates the acetylcholinesterase gene (*AcHE*), an important TLR regulator particularly relevant in neuroinflammation, which results in the inhibition of TLR-induced cytokine production [67]. The downregulation of the p300 transcriptional co-activator is a route utilized by Kaposi’s Sarcoma-associated Herpes Virus (KSHV) to evade host immunity through upregulating miR-132 expression. The resulting downregulation of p300 switches down antiviral genes interferon beta (*IFNβ*) and IFN-stimulated gene of 15 kDa (*IsG15*), and interleukin -1β (IL-1β) and -6 (IL-6), allowing the virus to propagate [68]. In addition to miR-132-3p, miR-590-5p is described as a pro-viral miRNA [69]. Zhou et al. [70] have shown that viral infections elevated miR-590-5p expression, and its increased level impaired antiviral pathways, which in turn promoted virus replication. MiR-590-5p attenuates the virus-induced expression of type 1 and type interferons and inflammatory cytokines, resulting in impaired antiviral signaling [70]. IL-6 receptor is a target of miR-590-5p. Similar to miR-132-3p, miR-590-5p was significantly deregulated in the serum of SSNHL patients [9]. These persisting miRNA changes in SSNHL patients may affect underlying innate immune responses to viral infection; however, further work is required to elucidate their relevance in SSNHL.

Circulating miRNAs have been shown to have high stability in the extracellular environment [71]. However, more work is required to optimize several key steps, such as miRNA extraction, normalization, quantitation, and analysis of serum miRNA profiles, before they can be used as reliable biomarkers of specific disease conditions. This is an initial exploratory study aimed at investigating the comparative expression levels of dysregulated serum miRNA in SSNHL patients over time. The conclusions are limited by the relatively small patient samples, the variation in the timing of sampling in the delayed group, and that identical patients were not studied in both groups. A further study is underway, which aims to address these concerns.

## 4. Materials and Methods

### 4.1. Study Populations and Sampling

Adult patients, 18 years and older, presenting with SSNHL as defined by the AAOHNS criteria [22], were recruited. Inclusion criteria included: age 18 years or older, signed informed consent to participate in the study, pure tone audiometrically documented sensorineural hearing loss of at least 30 dB across three contiguous frequencies that developed within 72 h from the onset, and presentation with symptoms of SSNHL within 28 days of onset. Patients were recruited whether or not they had undergone some form of treatment. 

Patients who were unable to provide any blood samples, complete diagnostic physical and hearing tests, or with any known cause for hearing loss, major intercurrent medical illness or coexisting ear pathologies were excluded from the study. 

### 4.2. Clinical Examination and Patient Recruitment

A full clinical examination of the participants’ ears was performed by the College of Physicians and Surgeons of British Columbia registered Otolaryngologist Head and Neck Surgeons. The position and shape of the external ears were inspected. Pain upon palpation over the tragus, pinna, or mastoid process was recorded if present. Otoscopic examination of the ear canals and tympanic membranes was undertaken to identify potential causes of conductive hearing loss resulting from middle ear fluid, otitis media, foreign bodies, cerumen impaction, otitis externa, or trauma. Webber’s tests were undertaken with 256 or 512Hz tuning forks to categorize hearing losses as sensorineural or conductive [72]. 

Patients with serum samples drawn 3-12 months after the onset of SSNHL (delayed group) were matched by age and sex to patients with samples drawn within 28 days of SSNHL onset (immediate group) (Table 2). This study was approved by the University of British Columbia’s Clinical Research Ethics Board, and 23 SSNHL patients were enrolled at the Department of Otolaryngology, Vancouver General Hospital, between 2017 and 2022.

### 4.3. Pure Tone Audiometry 

All participants underwent pure tone audiometric assessment by the College of Speech and Hearing Health Professionals of British Columbia registered Audiologists or Hearing Instrument Practitioners. The assessments were undertaken in sound-proofed environments using insert or supra-aural earphones and bone vibrators according to accepted practice guidelines [73]. Copies of the recorded results of these assessments were collected by the investigators and used to determine participant suitability for study enrollment. 

A sensorineural hearing loss was defined for study purposes as a loss of >25 dB at any frequency with an air-bone gap of <10 dB. A conductive hearing loss was defined as an air-bone gap of >25 dB at any frequency. A mixed hearing loss was defined as the occurrence of AC and BC losses of >25 dB, with an air-bone gap >10 dB [74,75]. During an audiometric assessment, if the patient does not respond to the maximum sound capacity of the audiometer, 5 dB was added to the maximum level to represent the threshold level. Only SSNHL patients with an averaged sensorineural hearing loss ≥30 dB across 3 contiguous frequencies were enrolled in the study. Control patients with any hearing loss >25 dB at any test frequency was excluded from the study. Any SSNHL patients with an associated conductive or mixed hearing loss were also excluded from the study. PTA audiometric thresholds across 4 low (0.5, 1, 2, and 3 or 4 kHz) or 3 high (3 or 4, 6, and 8 kHz) frequencies at presentation and post-treatment were used to determine hearing loss, recovery, and inter-group comparisons. The PTA threshold frequencies studied in each patient were based on the value consistent with the most severe hearing loss in the affected ear on presentation. 

### 4.4. Blood Collection and MicroRNA Extraction

Blood samples were collected for serum separation, and aliquots were stored at −80 °C for subsequent RNA extraction. Total RNA was extracted from 200 microliters of serum using miRNeasy Mini Kit (Qiagen, Toronto, ON, Canada) according to the manufacturer’s instructions. Briefly, 1 mL of Qiazol Lysis reagent was added to 200 μL of serum, followed by incubation at room temperature for 5 min. Then, 200 μL Chloroform (Sigma-Aldrich, St. Louis, MO, USA) was added, and the samples were vortexed and incubated for 5 min at room temperature followed by centrifugation at 12,000× *g* for 15 min at 4 °C. The aqueous phase containing RNA was transferred to a new tube, and isopropanol (Fisher-Scientific, Thermo Fisher Scientific, Waltham, MA, USA) was added. The samples were incubated at room temperature for 10 min followed by centrifugation at 12,000× *g* for 15 min at 4 °C. In total, 1.5 volumes of 100% ethanol were added and mixed thoroughly by pipetting. Up to 700 μL of the sample, including any precipitate, was pipetted into an RNeasy MinElute spin column in a 2 mL collection tube and centrifuge at ≥8000× *g* for 15 s at room temperature. Then, 700 μL Buffer RWT was added to the RNeasy MinElute spin column and centrifuge for 15 s at ≥8000× *g*. Then, 500 μL Buffer RPE was added onto the RNeasy MinElute spin column and centrifuge for 15 s at ≥8000× *g*. Then, 500 μL of 80% ethanol was added to the RNeasy MinElute spin column and centrifuged for 2 min at ≥8000× *g*. RNeasy MinElute spin column was placed in a new 2 mL collection tube with an open lid and centrifuged at full speed for 5 min to dry the membrane. After discarding the flow-through, 14 μL RNase-free water was added directly to the center of the spin column membrane.

### 4.5. MicroRNA Reverse Transcription

Reverse transcription (RT) was undertaken with a TaqMan Advanced cDNA synthesis kit (Applied Biosystems, Thermo Fisher Scientific, Waltham, MA, USA) utilizing a preamplification step followed by TaqMan™ advanced miRNA Assay (Applied Biosystems, Thermo Fisher Scientific, Waltham, MA, USA). In brief, a total of 30 μL and 2 μL of microRNAs, 3 μL of poly(A) reaction mix, 10 μL of ligation reaction mix, and 15 μL of reverse transcriptase reagents were incubated at 42 °C for 15 min and deactivated at 85 °C for 5 min. Then, 5 μL of the RT reaction product was incubated with 45 μL of the miR-Amp Reaction Mix before extracting and diluting cDNA 1:10 with nuclease-free water for microRNA real-time PCR analysis. 

### 4.6. MicroRNA Real-Time PCR

Real-time PCR was performed using Taqman^®^ Advanced miRNA Assays and Taqman Fast Advanced master mix on a QuantStudio Real-time PCR System (Applied Biosystems, Thermo Fisher Scientific, Waltham, MA, USA). Individual qPCR assays were performed in triplicate with a total reaction volume of 20 μL using hsa-miR-30a-3p/-128-3p/-132-3p/-140-3p/-186-5p/-195-5p -375-3p/-590-5p/primers according to the manufacturer’s instructions. The PCR thermocycler was programmed for denaturation at 95C for 20 s, followed by 40 annealing cycles of 95°C for 1 s and 60°C for 20 s. The normalization of microRNA expression levels was performed based on reference microRNA, hsa-miR-191-5p. The miRNA expression level was calculated using the delta Ct method [76].

### 4.7. Pure Tone Audiometry

Pure tone audiometric testing was used to confirm the diagnosis of SSNHL, categorize the degree of hearing loss at presentation, and determine patients’ hearing recovery status. Air conduction pure-tone average thresholds at 4 low (0.5,1,2 and 3 or 4 kHz) or 3 high (3 or 4,6 and 8 kHz) frequencies were calculated for affected ears [36]. The hearing threshold in the affected ear was used to classify the degree of SSNHL [77]. We selected the frequencies to be averaged in each patient (high or low) depending upon which frequencies demonstrated the poorest hearing at the initial pure tone audiogram. The same frequencies selected at the initial audiogram were averaged in all follow-up audiograms for each patient. The patients were also classified by hearing outcome into two groups, either hearing recovered or not recovered. The hearing recovered group was defined as patients whose PTA averaged hearing gain on follow-up was 10 dB or greater. The not recovered group displayed hearing gains of less than 10 dB [36].

### 4.8. Statistical Analysis

The expression level of the test miRNAs in the serum of SSNHL patients in the immediate and delayed groups was compared with Mann–Whitney U-tests using SPSS version 26 (IBM, Armonk, NY, USA). To determine if the groups were otherwise equally matched biologically inter-group, averaged age was statistically compared with independent samples. Student’s *t*-test or Welch’s *t*-test if the standard deviations in both groups were similar or different, respectively, and the inter-group proportion of females were compared with Pearson’s chi-square test. The groups’ patients’ clinical SSNHL profiles were compared statistically: initial and final PTA thresholds with Mann–Whitney U tests proportions of patients who recovered hearing and presented with associated dizziness were compared with Pearson’s chi-squared tests. A Bonferroni-corrected *p*-value of < 0.005 was adopted as significant to adjust for multiple t and u-tests. Box and Whisker plots were generated using GraphPad Prism version 9.0.

## Figures and Tables

**Figure 1 ijms-24-07307-f001:**
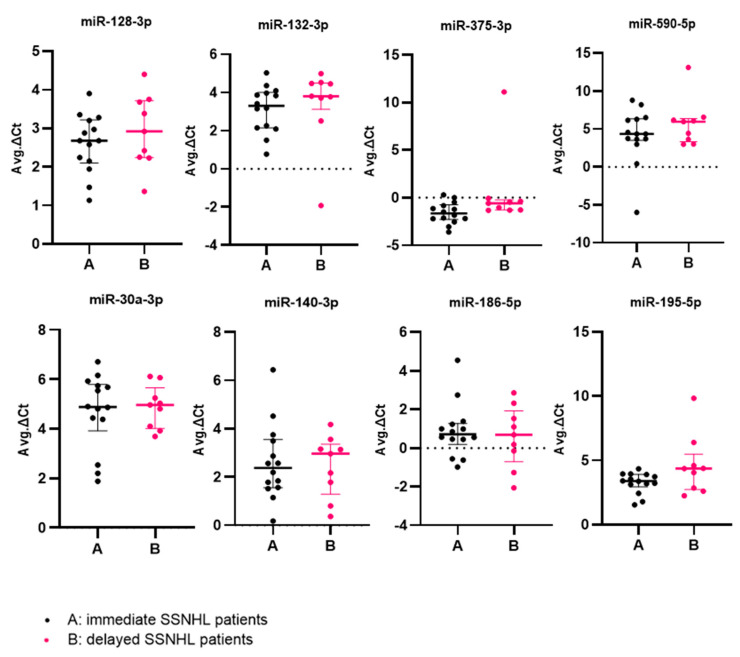
MicroRNA expression levels were the average of triplicate PCR runs for each miRNA in both groups. Box-plots of expression levels of miR-30a-3p, miR-128-3p, miR-132-3p, miR-375-5p, miR-590-5p, miR-140-3p, miR-186-5p, and miR-195-5p in sudden sensorineural hearing loss (SSNHL) immediate (A) and delayed (B) group patients. The central bar marks the median value, and the upper and lower bars, the 25th and 75th centile values. There was no significant difference in expression levels of these miRNAs between immediate and delayed group patients.

**Table 1 ijms-24-07307-t001:** Time of blood sampling from onset of the disease in delayed group patients.

Time of Blood Sampling from Onset of Hearing Loss in Delayed Patients	3 Months	6 Months	9 Months	12 Months
Number of Patients	2	3	2	2

**Table 2 ijms-24-07307-t002:** Characterization of sudden sensorineural hearing loss (SSNHL) in immediate and delayed sampled patients.

Demographic Details	Immediate Patients	Delayed Patients	*p* Value (Statistical Test)
Age in years (mean ± SD)	58.07 ± 16.95	59.56 ± 15.86	0.92 (independent samples *t*-test)
Sex (male:female)	7:7	5:4	0.36 (Pearson Chi square)
Dizziness (with dizziness:without dizziness)	0:14	3:6	0.11 (Chi square)
Hearing recovery (recovered:not recovered)	6:8 (42.9%:57.1%)	5:4 (55.6%:44.4%)	0.35 (Chi square)
Initial median PTA audiometric threshold of the affected ear (dB)	Q1: 40.1 Q3: 82.9 median: 66.9 min: 18.8 max:107.5	Q1: 58.5 Q3: 91.5 median: 76.3 min: 23.8 max: 105	0.33 (Mann–Whitney U-test)
Final median PTA audiometric threshold of the affected ear (dB)	Q1: 36.2 Q3: 79.1 median: 63.3 min: 8.8 max: 102.5	Q1: 14.4 Q3: 82.5 median: 67.5 min: 7.5 max: 105	0.92 (Mann–Whitney U-test)

## Data Availability

The data presented in this study are available upon request from the corresponding author.

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
