# Peer review of "The Expression Levels of MicroRNAs Differentially Expressed in Sudden Sensorineural Hearing Loss Patients’ Serum Are Unchanged for up to 12 Months after Hearing Loss Onset"

_ijms, 2023, doi:10.3390/ijms24087307_

Round 1

Reviewer 1 Report

The number of patients in the summary is not indicated.

Very few patients in groups 14 and 9. No difference or is there?

We don't know this because the size of the groups is not large enough to detect differences.

The disease can be caused by a number of reasons, and not taking this into account when forming groups is a serious mistake.

mean ± SD is used in the normal distribution. With such a number of patients, a normal distribution usually does not exist. 25, 50 (median), 75, as well as the minimum and maximum should be given. But even without this, it is clear that there is a wide variation in age, and, accordingly, the causes of the development of the disease. And also patients of different sexes. The studied groups are as heterogeneous as possible. How to compare them?

It is not clear why the authors indicated "willingness to provide blood samples at the time of enrollment and at 3 monthly intervals for 12 months". There is nothing about this in the results.

How much time has passed since the onset of the disease (at the time of blood sampling) in Delayed patients?

The title of the article does not match the content.

Reviewer 2 Report

Sudden sensorineural hearing loss (SSNHL) is a type of acquired idiopathic hearing loss developed within 3 days and has an incurrence rate around 0.03%. Existing evidence suggested involvement of miRNA in SSNHL and a number of miRNAs have been reported to be differentially expressed in the blood of SSNHL patients in the early stage, but it is unknown whether the changes in these miRNAs persist in the later stage. In this study, Abgoon and colleagues compared the expression level of selected 8 miRNAs in SSNHL patients within 1 month of hearing loss onset and those patients at 3-12 months stage. They detected no significant inter-group difference in the miRNA expression level, which highlights the persistence of differentially expressed miRNAs in SSNHL patients. This is a straightforward and meaningful clinic investigation. The data is very net and supports the author’s conclusion. I only have some minor comments below.

1.       Line 21: add the abbreviation "PTA" here to "pure-tone audiometric" since it was used in the later part of the abstract.

2.       Line 31: should be "pre-miRNAs", not "pri-miRNAs".

3.       I strongly suggest the author to discuss the limitation of their study in the end, i.e., an improved investigation design would assess the miRNA level in the SSNHL patients with multiple sample collections from early stage to later stage and compared the expression changes for each individual over time. I understand this could be difficult practically, but it is better to discuss in a bit detail.

Round 2

Reviewer 1 Report

The authors corrected a significant part of the comments. The authors indicated the limitations of the study. Of course, the meager size of the subjects did not become larger from this. The authors indicated the group sizes explicitly in the abstract. This will enable the reader to independently decide whether it is worth spending time reading an article about a study performed on such material.